# Changes in Apolipoprotein A1-Associated Proteomic Composition After Pioglitazone Treatment Versus Weight Loss

**DOI:** 10.3390/ijms262110690

**Published:** 2025-11-03

**Authors:** Shyon Parsa, Timothy S. Collier, Michael J. McPhaul, Olle Melander, Joshua W. Knowles, Anand Rohatgi, Fahim Abbasi

**Affiliations:** 1Diabetes Research Center, Division of Cardiovascular Medicine and Cardiovascular Institute, School of Medicine, Stanford University, Stanford, CA 94305, USA; knowlej@stanford.edu (J.W.K.); fahim@stanford.edu (F.A.); 2Quest Cardiometabolic Center of Excellence at Heartlab, Inc., Cleveland, OH 44103, USA; timothy.s.collier@questdiagnostics.com; 3Quest Diagnostics Nichols Institute, San Juan Capistrano, CA 33608, USA; michael.mcphaul@utsouthwestern.edu; 4Division of Endocrinology, UT Southwestern Department of Medicine, Dallas, TX 75390, USA; 5Department of Clinical Sciences Malmö, Lund University, 22242 Lund, Sweden; olle.melander@med.lu.se; 6Department of Internal Medicine, Skane University Hospital, 20213 Malmö, Sweden; 7Division of Cardiovascular Medicine, UT Southwestern Department of Medicine, Dallas, TX 75390, USA; anand.rohatgi@utsouthwestern.edu

**Keywords:** insulin resistance, apolipoprotein A1, proteomics, atherogenic dyslipidemia, pioglitazone

## Abstract

Insulin resistance (IR) contributes to atherogenic dyslipidemia and elevated ASCVD risk. Apolipoprotein A1 (ApoA1)-associated lipoproteins have diverse anti-atherogenic functions, but it is unclear whether IR drives adverse changes in their proteomic composition. We hypothesized that IR is associated with an atherogenic ApoA1 proteome and that insulin-sensitizing interventions would improve its composition. We studied 861 participants without diabetes (age 47 ± 12 years, 65.5% female). IR was directly measured using the steady-state plasma glucose (SSPG) concentration via the insulin suppression test. ApoA1-associated proteins were quantified by mass spectrometry. A subset underwent interventions for 3 months (N total 108): pioglitazone, PIO *n* = 38 or weight loss, WL *n* = 70). Paired t-tests assessed pre- and post-intervention changes. At baseline, several ApoA1-associated proteins significantly correlated with SSPG. Both interventions improved IR (*p* < 0.01). PIO led to significant increases in 14 ApoA1-associated proteins, including ApoC1–C4, ApoA2, ApoA4, ApoD, ApoE, LCAT, and PON1/3. WL increased several ApoA1-associated proteins, including ApoA4, ApoD, ApoM, and PON1/3. In conclusion, IR is associated with a pro-atherogenic ApoA1 proteome, and both interventions improve this profile. However, PIO has a broader proteomic impact. These findings highlight the potential of targeting the ApoA1 proteome to reduce residual ASCVD risk.

## 1. Introduction

Cardiometabolic diseases, including atherosclerotic cardiovascular disease (ASCVD) and type 2 diabetes mellitus (T2DM), remain among the leading causes of morbidity and mortality worldwide [1,2]. Insulin resistance (IR) is a hallmark of cardiometabolic dysfunction and has been linked to an atherogenic lipid profile [3,4]. This dyslipidemia pattern is characterized by elevated triglycerides, small dense low-density lipoprotein (LDL) particles, and reduced high-density lipoprotein (HDL) cholesterol levels [5]. Individuals with increased IR have heightened ASCVD risk, and a proportion of their residual risk remains inadequately addressed by conventional lipid-lowering therapies [6,7].

High-density lipoproteins (HDL) contain more than 200 proteins, with the most abundant being apolipoprotein A1 (ApoA1) [8,9]. HDL exerts cardioprotective effects primarily through reverse cholesterol transport (RCT), which facilitates the removal of cholesterol from macrophages in atherosclerotic plaques to the liver for excretion [10]. 

Additionally, HDL particles reduce oxidative stress and inflammatory responses by neutralizing reactive oxygen species and inhibiting monocyte adhesion and cytokine release [11]. ApoA1 is mechanistically linked to enhanced insulin sensitivity, β-cell function, and RCT, highlighting its role as a molecular link between IR, T2DM, and ASCVD [12,13]. However, crude measures of HDL cholesterol or ApoA1 levels inadequately capture the functional heterogeneity of HDL particles [14]. Advanced proteomic studies reveal that ApoA1 interacts with a diverse array of associated proteins, collectively termed the ApoA-I-associated proteome [15,16]. These proteins contribute to varied—and sometimes opposing—effects on lipid metabolism, inflammation, and vascular health, highlighting the importance of characterizing the individual and collective roles of ApoA-I-associated proteins in ASCVD risk stratification.

IR is not only a precursor to T2DM but also drives an atherogenic state that may adversely remodel the ApoA1 proteome [17]. Emerging evidence suggests that individuals with IR exhibit altered ApoA1 proteomic profiles, potentially diminishing HDL functionality and exacerbating ASCVD risk [18,19]. Prior work using high-throughput techniques identified a combination of five apolipoproteins (ApoA1, C1, C2, C3, and C4) to develop a score known as pCAD that correlated with prevalent obstructive coronary artery disease (CAD) [20]. This correlation was independent of traditional risk factors, ApoA-I, and ApoB levels and served as a marker for those with a more atherogenic ASCVD risk profile. Moreover, in individuals without ASCVD history, the pCAD score independently predicted incident ASCVD events [21]. It remains unknown whether IR severity correlates with specific changes in the ApoA1 proteome—including pCAD—and whether targeted improvements in IR can reverse these atherogenic proteomic changes.

Therapeutic strategies aimed at improving insulin sensitivity, such as pioglitazone (PIO), a peroxisome proliferator-activated receptor-γ (PPAR-γ) agonist, and lifestyle interventions like weight loss (WL), have shown promise in mitigating cardiometabolic risk [22,23,24]. PIO enhances insulin sensitivity by activating PPAR-γ, which regulates gene expression involved in glucose and lipid metabolism, thereby improving adipocyte function and reducing systemic inflammation [25,26]. In contrast, WL primarily reduces IR by decreasing adipose tissue mass, leading to lower levels of circulating free fatty acids and inflammatory cytokines that impair insulin signaling [27]. However, their specific effects on the ApoA1 proteome remain underexplored. By comparing these two interventions, we aimed to isolate the proteomic effects driven by PPAR-γ–mediated transcriptional remodeling from those due to adipose reduction alone. Understanding these effects may not only elucidate the pathophysiology linking IR to ASCVD but also provide novel biomarkers or therapeutic targets for reducing residual ASCVD risk.

In this study, we tested the hypothesis that IR is associated with an atherogenic ApoA1 proteome and that insulin-sensitizing interventions can remodel the proteome to a more anti-atherogenic composition. This hypothesis directly addresses gaps in understanding the relationship between IR and the functional diversity of ApoA-I-associated proteins—specifically how IR exacerbates atherogenic proteomic changes and whether therapeutic strategies can reverse these alterations. Using direct measures of IR via steady-state plasma glucose (SSPG) concentration during the insulin suppression test, we characterized the ApoA1 proteome in a cohort of individuals without diabetes and assessed the effects of PIO and WL interventions. By exploring the interplay between IR and the ApoA1 proteome, this study aims to advance our understanding of residual lipid-related ASCVD risk and identify potential therapeutic strategies to mitigate it.

## 2. Results

### 2.1. Baseline Characteristics

The study cohort consisted of 861 participants (age 47 ± 12 years, 65.5% female) without diabetes who participated in the Studies of Insulin Resistance at Stanford (Table 1). Baseline SSPG concentrations were positively correlated with markers of IR, including IRRS (r = 0.73, *p* < 0.001) and HOMA-IR (r = 0.67, *p* < 0.001) (Table 2).

### 2.2. Baseline Correlations Between IRRS and SSPG and ApoA-I-Associated Proteome

At baseline, several ApoA-I-associated proteins significantly correlated with SSPG concentrations (Figure 1). Negative correlations were observed for multiple ApoA1-associated proteins, including ApoA1 (r = −0.27, *p* < 0.001) and ApoC1 (r = −0.18, *p* = 0.001), both associated with an atherogenic profile. Conversely, ApoL1 was positively correlated with SSPG concentration (r = 0.20, *p* < 0.001). IRRS and ApoA1 proteome were measured in 858 individuals without DM (Studies of Insulin Resistance at Stanford), revealing correlations between IRRS and multiple proteomic markers and consistent correlations compared to the gold-standard SSPG concentration measurement of IR (Figure 2). These data support the correlative nature of IR with the ApoA1 proteome and the noninvasive IRRS method as a reflection of the gold-standard assessment of IR (SSPG concentration).

### 2.3. Intervention Effects on Insulin Resistance and Lipid Levels

Of the 861 participants with measured IRRS and ApoA1 proteomic composition, a subgroup of participants underwent either pioglitazone treatment (PIO; *n* = 38) or weight loss (WL; *n* = 70) for 3 months, when all baseline measurements were repeated (Appendix A). Participants were initially recruited through media advertisements, and the PIO and WL studies were separate, with results presented together. For both studies, participants were included if they did not have a diagnosis of diabetes, were not taking glucose-lowering medication, and had a plasma glucose concentration of less than 126 mg/dL on the screening visit. For the weight loss studies, participants were included if their body mass index (BMI) was between 25 kg/m^2^ and 40.0 kg/m^2^. Among the interventional study cohort, those in the PIO arm had a lower baseline weight, lower BMI, and lower fasting insulin concentrations (Table 3). Both pioglitazone (PIO) and weight loss (WL) interventions significantly improved SSPG concentrations after 3 months (Appendix A). PIO reduced SSPG concentration by 30.4% (baseline: 163 ± 67 mg/dL to post-intervention: 113 ± 55 mg/dL, *p* < 0.01). WL reduced SSPG concentration by 18.2% (baseline: 158 ± 72 mg/dL to post-intervention: 129 ± 60 mg/dL, *p* < 0.01). The reduction in SSPG concentration was significantly greater with PIO compared to WL (*p* = 0.03). However, there was a small increase in weight in the PIO group with a mean (SD) of 0.9 kg (2.4 kg). In addition, after PIO, there were reductions in total cholesterol, triglycerides, and LDL-C levels; however, these differences did not reach statistical significance (Appendix A). After WL, there were significant reductions in total cholesterol, triglycerides, and LDL-C levels. HDL-C levels were not statistically different after either intervention.

### 2.4. Changes in ApoA-I-Associated Proteome

PIO treatment led to significant increases in several ApoA1-associated proteins: ApoC1, ApoC2, ApoC3, ApoC4, ApoA2, ApoA4, ApoD, ApoE, ApoM, LCAT, LpPLA2, PLTP, PON1, and PON3. (Figure 3a, Appendix A). WL also led to significant increases in ApoA1-associated proteins: ANGT, ApoA4, ApoD, ApoM, LpPLA2, PLTP, PON1, and PON3 (Figure 3b, Appendix A). The calculated pCAD index decreased from a mean of 64.2 (SD 32.4) to 54.8 (SD 37.0) in the PIO arm; however, this change was not statistically significant (% change = 14.6%, *p* = 0.08). The calculated pCAD index decreased from a mean of 75.6 (SD 27.1) to 72.8 (SD 28.1) in the WL, which was a nonsignificant result (% change = 3.7%, *p* = 0.46). An analysis of the relationships between the IR marker IRRS and the ApoA1 proteome revealed that among the PIO intervention group, the change in IRRS was significantly associated with the change in ApoE, CO3, LpPLA2, PTLP, and TTHY. Among the WL cohort, the change in SSPG concentration was significantly associated with ApoA4 and ApoD. The change in IRRS was associated with CLUS and KAIN.

In this study of 861 individuals without diabetes, we found that insulin resistance (IR), as measured by SSPG and IRRS, was significantly associated with an atherogenic ApoA-I-associated proteome. Both pioglitazone (PIO) and weight loss (WL) interventions improved insulin sensitivity, but PIO induced a broader and more pronounced remodeling of the ApoA1 proteome, including significant increases in pCAD-related proteins (ApoC1, ApoC2, ApoC3, and ApoC4), as well as anti-inflammatory markers such as PON1 and PON3. In contrast, WL was associated with smaller proteomic changes and no significant improvement in pCAD proteins.

Our findings provide novel insights into the relationship between IR and the ApoA-I-associated proteome and the potential for targeted interventions to mitigate ASCVD risk. The positive correlation between SSPG concentration and atherogenic proteins highlights the dysregulation of HDL functionality in the context of IR. The comparative analysis of PIO and WL reveals nuanced differences between the mechanisms of HDL remodeling among therapeutics, and specifically for ApoA-I-associated proteins. Our study supports previous work and highlights pathophysiological alterations of lipid metabolism in metabolic dysfunction.

The primary limitations of prior lipoprotein proteomic studies have been the difficulties in quantifying individual proteins, which have demonstrated insufficient predictive capacity for ASCVD risk beyond traditional lipid markers and established risk factors [28,29]. Our investigation provides a comprehensive characterization of the ApoA1 proteomic signature by identifying multiple proteins using a scalable analytical platform. By analyzing multiple ApoA-I-associated proteins simultaneously, we provide a more nuanced understanding of HDL functionality that extends beyond traditional single-protein measurements.

In addition, previous research on IR assessment has primarily utilized indirect measurement techniques that prioritize ease of use, broad applicability, and cost-effectiveness [30,31]. These indirect measurements, such as the homeostasis model assessment of insulin resistance (HOMA-IR) and those derived during an oral glucose tolerance test, are typically only modestly correlated with direct measures of IR. Furthermore, these indirect measurements have known inconsistencies due to a lack of insulin assay standardization [32,33,34]. Gold-standard direct techniques for measurement of the degree of IR, such as the insulin suppression test and euglycemic hyperinsulinemic clamp, are invasive and labor-intensive [35,36,37]. Our study leverages the IRRS with mass-based measures, which demonstrated excellent correlation and calibration against the gold-standard SSPG concentration measurement [38]. This approach creates a simpler and more cost-effective framework for future studies aiming to assess IR in larger population cohorts.

This study is the first to investigate the association of ApoA1 proteomic signatures in healthy individuals who are free of ASCVD and DM in a large cohort (i.e., in the primary prevention setting). PIO and WL both improved IR, as evidenced by significant reductions in SSPG concentrations. However, the proteomic changes observed as a result of these interventions highlight the underlying differences between pharmacologic and lifestyle-based strategies. PIO demonstrated a more pronounced effect on restoring an anti-atherogenic ApoA1 proteome, including significant increases in PON1 and pCAD protein levels, such as ApoC1, ApoC2, ApoC3, and ApoC4, independent of weight loss. This supports previous work that PIO may modulate proteomic composition through anti-inflammatory and lipid-regulating pathways, consistent with its known activation of peroxisome proliferator-activated receptor-γ (PPAR-γ) [39]. PPAR-γ activation has been shown to alter gene expression in liver and adipose tissue, lipase activity, and lipoprotein remodeling [39]. In contrast, WL led to modest improvements in proteomic markers such as PON1 and PON3, reflecting the benefits of reduced adipose tissue and associated pro-inflammatory mediators. Of note, changes in response to pioglitazone are temporary if the medication is withdrawn, while changes related to weight loss tend to persist if the weight loss is maintained. However, the lack of significant changes in pCAD-associated proteins reveals the limitations of WL alone in reversing HDL dysfunction in insulin-resistant individuals [40]. This may be a result of a lack of influence from the interventions in the study or the necessity for a larger sample size to detect changes in the pCAD index. Studies suggest that larger degrees of weight loss may be required to observe changes in HDL functionality [41,42].

The comparative analysis of PIO and WL reveals key insights into the therapeutic potential of targeting the ApoA-I-associated proteome. The greater improvement in composite proteomic scores with PIO suggests its unique capacity to remodel HDL functionality, particularly in individuals with higher degrees of insulin resistance. Our work provides a potential mechanism for previous studies that have outlined the beneficial effects of PIO specifically in those with known IR [43,44]. The following studies have also supported downstream changes as a result of PIO use. One work that examined carotid artery intimal thickness (CIMT), a known surrogate of atherosclerosis, before and after PIO use discovered that increases in HDL cholesterol served as an independent predictor of slowing CIMT progression within subjects in the PIO arm [45]. The RADIANCE-2 study found that while torcetrapib significantly increased HDL cholesterol, progression of CIMT was unchanged in the torcetrapib arm versus statin alone [46]. This finding suggests that the effect of PIO on ASCVD risk is multifactorial rather than solely a result of increased HDL cholesterol. In this context, our study supports that remodeling the ApoA1 proteome in IR is one pathway that warrants further investigation.

From a clinical perspective, these findings raise intriguing questions about the potential of ApoA1 proteomic profiling as a biomarker for stratifying ASCVD risk or guiding therapy. Longitudinal studies with cardiovascular outcomes are needed to assess whether remodeling of the HDL proteome corresponds with reductions in major adverse events. Moreover, as pioglitazone carries known safety concerns, future translational studies must weigh these risks against any potential benefit from HDL-targeted therapies. Ultimately, integrating functional validation with pharmacologic safety profiles will be essential to determine whether modifying the HDL proteome can meaningfully reduce ASCVD burden.

The observed improvements in the ApoA1-associated proteome following pioglitazone are biologically plausible given the well-established roles of specific apolipoproteins in lipid metabolism. For instance, ApoC3 is a potent inhibitor of lipoprotein lipase (LPL) and hepatic lipase, delaying triglyceride-rich lipoprotein clearance [47]. ApoE, which increased significantly with pioglitazone, serves as a ligand for hepatic remnant receptor-mediated clearance, and its enrichment in HDL particles may improve reverse cholesterol transport and remnant lipoprotein clearance [48]. Of note, it is likely that PPAR-gamma agonists affect the clock genes (e.g., BMAL1) and may contribute to shifting peaks in LACT or ApoE levels. Similarly, elevated levels of ANGPTL3 and ANGT have been shown to inhibit LPL activity, and reductions in their levels are expected to derepress LPL and promote triglyceride hydrolysis [49]. Increases in proteins like ApoM and PON1/3, which are involved in HDL stability, antioxidant capacity, and sphingosine-1-phosphate (S1P) transport, may further reflect a shift toward a more anti-atherogenic HDL phenotype [50,51]. These findings suggest that the proteomic remodeling of HDL by insulin-sensitizing interventions is mechanistically aligned with improved lipid handling and atherosclerotic risk reduction.

This study has several strengths, including the use of a gold-standard measure of IR and a robust proteomic platform. However, some limitations should be noted. The sample size for the PIO group was relatively small, and the open-label design may introduce selection bias. Additionally, the generalizability of findings to diverse populations requires further validation. Despite these limitations, our findings underscore the importance of addressing HDL dysfunction in insulin-resistant individuals and provide a foundation for studies targeting the ApoA1 proteome as a therapeutic goal. Future research should explore the long-term clinical implications of these proteomic changes, including their association with incident cardiovascular events.

## 3. Materials and Methods

### 3.1. Study Design and Participants

This study included 861 participants without diabetes who were recruited as part of the Studies of Insulin Resistance at Stanford. Participants provided written informed consent, and the study protocol was approved by the institutional review board at Stanford University (11987 (20-December-2001) and 13091 (20-February-2001)). Inclusion criteria required participants to be free of diabetes mellitus, confirmed by fasting plasma glucose levels < 126 mg/dL and the absence of glucose-lowering medications. Key exclusion criteria included significant comorbidities or recent cardiovascular events.

### 3.2. Assessment of Insulin Resistance

IR was directly measured using the steady-state plasma glucose (SSPG) concentration during the insulin suppression test, a gold-standard method for quantifying insulin-stimulated glucose uptake [52]. The same technician and laboratory were used to process and analyze all insulin suppression tests. Participants underwent the test after an overnight fast. Two intravenous catheters were placed, one in each arm. One arm delivered a continuous infusion of octreotide acetate (0.27 μg/m^2^/min), insulin (32 mU/m^2^/min), and glucose (267 mg/m^2^/min) for 180 min. Blood samples were collected from the opposite arm at 30-min intervals up to 150 min and every 10 min thereafter. SSPG values were determined by averaging glucose concentrations from samples collected between 150 and 180 min, where higher values indicated greater IR. Baseline SSPG concentration median (132.5 mg/dL) was used to define subjects as being insulin resistant (SSPG > 132.5 mg/dL) or insulin sensitive (SSPG ≤ 132.5 mg/dL). The Homeostasis Assessment of Insulin Resistance (HOMA-IR) was calculated for participants based on previously described methods [53].

The insulin resistance risk score (IRRS) was calculated using the following equations as described previously [38]:RS = (insulin × 0.0295) + (C-peptide × 0.00372)
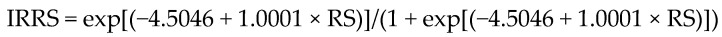


Insulin and C-peptide levels were measured by mass spectrometry assay in baseline fasting serum samples stored at −80 °C and standardized to the peptide content of WHO reference materials [54]. The IRRS is reported on a probability scale (0–100%).

### 3.3. Measurement of ApoA-I-Associated Proteome

The ApoA-I-associated proteome was characterized using a previously described and validated method of high-throughput mass spectrometry at the Quest Diagnostics Cleveland HeartLab (Cleveland, OH, USA) [15]. The samples for the ApoA1 protein were collected on the same morning of the Insulin Suppression Test (IST) using a standardized method that was applied to all study participants. Plasma samples were collected at baseline and, for participants in the intervention groups, at 3 months post-intervention. ApoA-I-associated proteins were enriched using an affinity-based protocol employing recombinant ApoA-I, which isolates proteins interacting with ApoA-I-containing HDL particles. Proteins were then analyzed using liquid chromatography-tandem mass spectrometry (LC-MS/MS). Both targeted and untargeted approaches were employed. In the targeted proteomics a focused analysis of 28 validated ApoA-I-associated proteins included key components such as ApoA1, ApoA2, ApoA4, ApoC1, ApoC2, ApoC3, ApoC4, ApoD, ApoE, ApoL1, ApoM, Alpha-1-antitrypsin (A1AT), angiotensinogen (ANGT), cholesteryl ester transfer protein (CETP), clustering (CLUS), complement-CIII (CO3), haptoglobin (Hp), kallistatin (KAIN), lecithin cholesterol acyltransferase (LCAT), lipoprotein-associated phospholipase-A2 (LpPLA2), phospholipid transfer protein (PLTP), serum paraoxonase/arylesterase-1 (PON1), serum paraoxonase/arylesterase-3 (PON3), retinol-binding protein-4 (RET4), serum amyloid alpha -1/2 (SAA1/2), serum amyloid alpha -4 (SAA4), transthyretin (TTHY), and vitronectin (VTN). Untargeted discovery consisted of comprehensive profiling of all ApoA-I-associated proteins to identify novel candidates associated with IR and ASCVD risk. The prediction of CAD risk (pCAD) algorithm contains a linear combination of five apolipoproteins (ApoA1, C1, C2, C3, and C4) from which the prediction of CAD risk is identified [55,56]. A pCAD score was calculated for study participants. Proteins were quantified relative to isotope-labeled standards, and coefficients of variation (CV) for repeat measurements ranged from 5% to 14%, ensuring reliability.

### 3.4. Sample Preparation

Specimens were prepared according to a CAP/CLIA-validated laboratory-developed test (LDT) protocol as previously described [55]. Batches were prepared in 96-well plates using a FreedomEvo v2.4 automated liquid handler utilizing Freedom EVOware v2.4 (Tecan Group). Specimen serum was combined with 15N-labeled, N-terminally His-tagged apolipoprotein A1 (15N-His6ApoA-I, Proteos, Kalamazoo, MI, USA) and incubated at 37 °C for 20 min, during which the labeled ApoA1 freely associated with HDL in the serum specimen. HDL-associated proteins were subsequently bound, washed, and eluted from tip-based Ni-NTA columns (Phytips, Biotage, San Jose, CA, USA). Eluted specimens were then subjected to heat denaturation followed by digestion by endoproteinase LysC (Santa Cruz Biotechnology, Dallas, TX, USA) and internal standard peptide addition.

### 3.5. LC-MS Parameters

For pCAD-associated proteins (apolipoproteins A1, C1, C2, C3, and C4), liquid chromatographic separation was performed according to the CAP/CLIA-validated LDT protocol as previously described [55]. Separation was performed on an Agilent 1260 Infinity LC system (Agilent Technologies, Santa Clara, CA, USA) using StreamSelect Software v.8.0.8023 to allow the operation of multiplexed LC channels. After a 10 µL injection, peptides were eluted using a gradient consisting of 2% mobile phase B (MPB, 0.1% formic acid in acetonitrile; MPA, 0.1% formic acid in water) for 1.25 min, ramping to 35% MPB by 4.75 min, ramping to 95% MPB in 0.1 min, and holding until 7.25 min before equilibrating at 2% MPB for an additional 1.75 min for a total run-time of 9 min. All peptides eluted within a 3-min window, allowing for the multiplexing of 3 LCs at 100% duty cycle to speed batch analyses.

For remaining protein targets, a second 10 µL injection from each specimen was performed, and peptides were eluted according to a gradient consisting of 2% MPB for 1 min, ramping to 13% MPB by 3 min, then to 30% MPB by 10 min, before increasing to 95% MPB at 10.1 min and holding at 95% MPB until 12 min before equilibrating at 2% MPB again for the remaining LC time for a total time of 15 min. For both LC gradients, eluted peptides were detected using an Agilent 6495C mass spectrometer (Agilent Technologies, Santa Clara, CA, USA), utilizing Stream Select v.8.0.8023, operating in positive ion mode with an electrospray source operating at a nebulizer gas temperature of 250 °C, a nebulizer gas flow of 17 L/min, 30 psi nebulizer pressure, with sheath gas heated to 250 °C flowing at 12 L/min, and a capillary voltage of 3000 V.

The mass spectrometer utilized a targeted dynamic MRM method. Eluted peptides were detected on an Agilent 6495C triple quadrupole mass spectrometer operating in dynamic multiple reaction monitoring mode. Targeted proteins were quantified by isotope dilution with the addition of isotope-diluted internal standards for each peptide containing stable isotope-labeled C-terminal lysine residues and assigned concentrations by triplicate amino acid analysis to allow absolute quantification. Two peptides for each protein were targeted, with two MRM transitions measured per peptide. Reported protein quantities in nanomoles per liter (nM) are based on the peptide transition with the most abundant MS response. Appendix A describes the targeted proteins, peptide sequences, primary and secondary MRM ion transition pairs, internal standard MRM transitions, assigned calibrant values, and validated analytical measurement ranges where available. In the absence of a defined measurement range, reported values were accepted if quality data met acceptance criteria (peptide and IS retention time alignment and peptide and/or ion ratios within 20% of target values).

Quality of data across multiple batches was ensured through multiple approaches. Quality of targeted MS data was tracked using the ratio of quantifier and qualifier transitions for each peptide, in addition to the ratio of the reported quantities of the 2 peptides targeted for each protein. Batch-to-batch quality was also tracked with the inclusion of three levels of quality control specimens with known target values bracketing unknowns in each batch and evaluated using Westgard rules. Apart from a specimen ID number, the analytical laboratory was blinded to any additional specimen-associated information. Specimens were randomized across batches.

### 3.6. Intervention Groups

A subgroup of participants underwent one of two insulin-sensitizing interventions: pioglitazone (PIO) treatment or a structured weight loss (WL) program. In the pioglitazone protocol, PIO was initiated at 15 mg daily for the first 2 weeks, increased to 30 mg daily for the next 2 weeks, and then maintained at 45 mg daily for the remaining 8 weeks. Participants in the WL program engaged in a calorie-restricted diet that included a 500–750 kcal deficit, guided by a registered dietitian, with a target of losing 0.5 kg per week. Weekly follow-ups were conducted to ensure adherence and provide counseling. Both interventions were designed to improve insulin sensitivity, with all measurements repeated at the 3-month follow-up.

### 3.7. Outcome Measures

The primary outcome of interest was the relationship between SSPG concentration and changes in ApoA-I-associated proteome composition. Secondary outcomes included changes in weight, fasting lipid profiles, and correlations between specific ApoA-I-associated proteins and SSPG concentration. For the intervention groups, pre- and post-treatment changes in both SSPG concentration and proteomic composition were analyzed.

### 3.8. Statistical Analysis

Descriptive statistics were used to summarize baseline characteristics, with continuous variables reported as means (± standard deviations) and categorical variables as proportions. Correlations between SSPG concentration and specific ApoA-I-associated proteins, assessed using Pearson correlation coefficients. Paired t-tests were employed to compare pre- and post-treatment changes within each intervention group. Between-group differences (PIO vs. WL) were analyzed using independent *t*-tests. To account for multiple hypothesis testing, the Benjamini–Hochberg procedure was applied, with a false discovery rate set at 5%. Statistical significance was defined as *p* < 0.05. All analyses were performed using R: A Language and Environment for Statistical Computing version 4.3.3 (R Foundation for Statistical Computing, Vienna, Austria).

To evaluate the relationship between protein profiles in participants stratified by insulin resistance status and the ApoA1 proteome, we calculated Pearson correlation coefficients using the pandas library (version 1.4.4). This analysis quantified the linear associations between protein expression levels before and after the cardiometabolic intervention within each group. For each protein pair, the correlation analysis was conducted independently before and after cardiometabolic intervention. Missing values were excluded from the correlation analysis. Proteins with statistically significant correlations (*p* < 0.05) were labeled.

### 3.9. Power Calculations

Sample size calculations indicated that 38 participants in the PIO group and 70 in the WL group provided 80% power to detect a 0.5 standard deviation change in SSPG concentration or ApoA1 protein levels (α = 0.05). These estimates were based on anticipated changes in SSPG concentration derived from prior studies and validated proteomic methods.

### 3.10. Data Handling and Missing Values

Missing data were assumed to be missing at random. Characteristics of participants with incomplete data were evaluated, and multiple imputation was performed where necessary. Sensitivity analyses were conducted to ensure robustness. Advanced imputation methods tailored to complex data structures to handle missingness effectively were applied.

## 4. Conclusions

Our work highlights the important role of IR in altering the ApoA-I-associated proteome and the potential of therapeutic interventions to modulate this profile. PIO demonstrated superior effects in restoring an anti-atherogenic proteomic composition compared to weight loss, emphasizing the therapeutics’ potential in addressing residual cardiovascular risk, particularly in insulin-resistant individuals. Future studies should investigate the long-term clinical impact of these proteomic changes and explore personalized treatment strategies tailored to the severity of IR. By advancing our understanding of HDL functionality, this research provides the basis for studying and testing novel therapeutic approaches for reducing cardiometabolic risk.

## Figures and Tables

**Figure 1 ijms-26-10690-f001:**
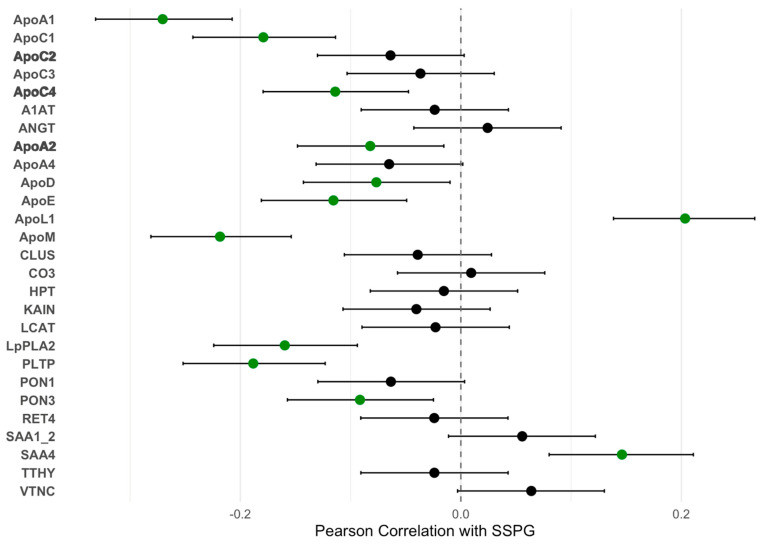
Correlations between ApoA1-Associated Proteins with IR [SSPG concentration]. SSPG indicates steady-state plasma glucose and is a direct measure of insulin resistance (IR) obtained during the insulin suppression test. A higher SSPG concentration indicates greater IR. Filled circles represent the Pearson correlation coefficients (r values), and horizontal error bars represent the 95% confidence intervals of the correlation coefficients. Green circles represent a statistically significant association (*p* < 0.05). **Bold text** indicates changes in ApoA1-associated proteins that result in an improved atherogenic profile.

**Figure 2 ijms-26-10690-f002:**
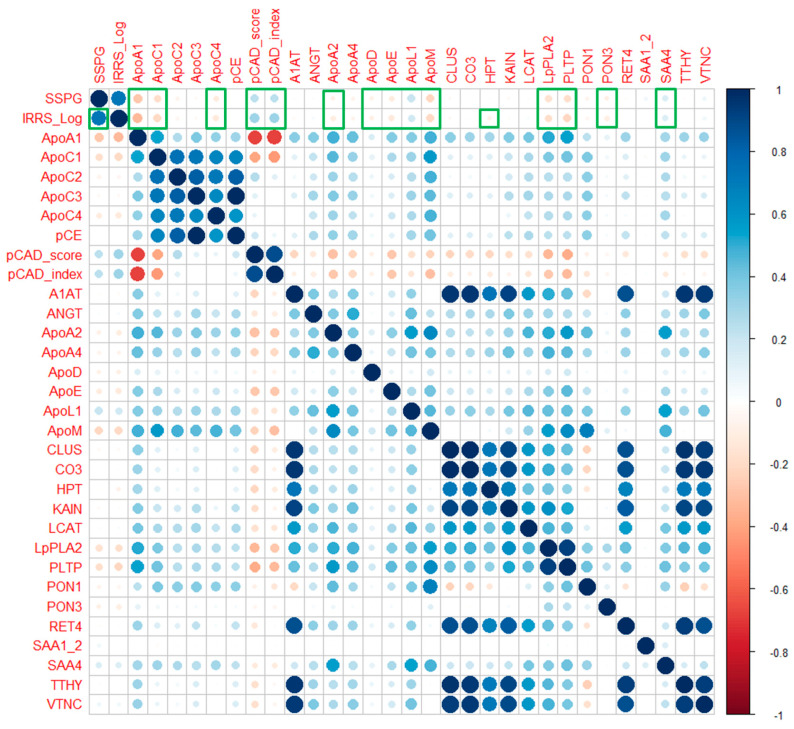
Heatmap showing heterogeneity between baseline SSPG concentration and IRRS with ApoA1 proteome. (N = 858). Correlation between IRRS and SSPG (insulin resistance) and ApoA1 proteomic markers. Color intensity and size are proportional to strength. Green outlines highlight statistically significant correlations (*p* < 0.05).

**Figure 3 ijms-26-10690-f003:**
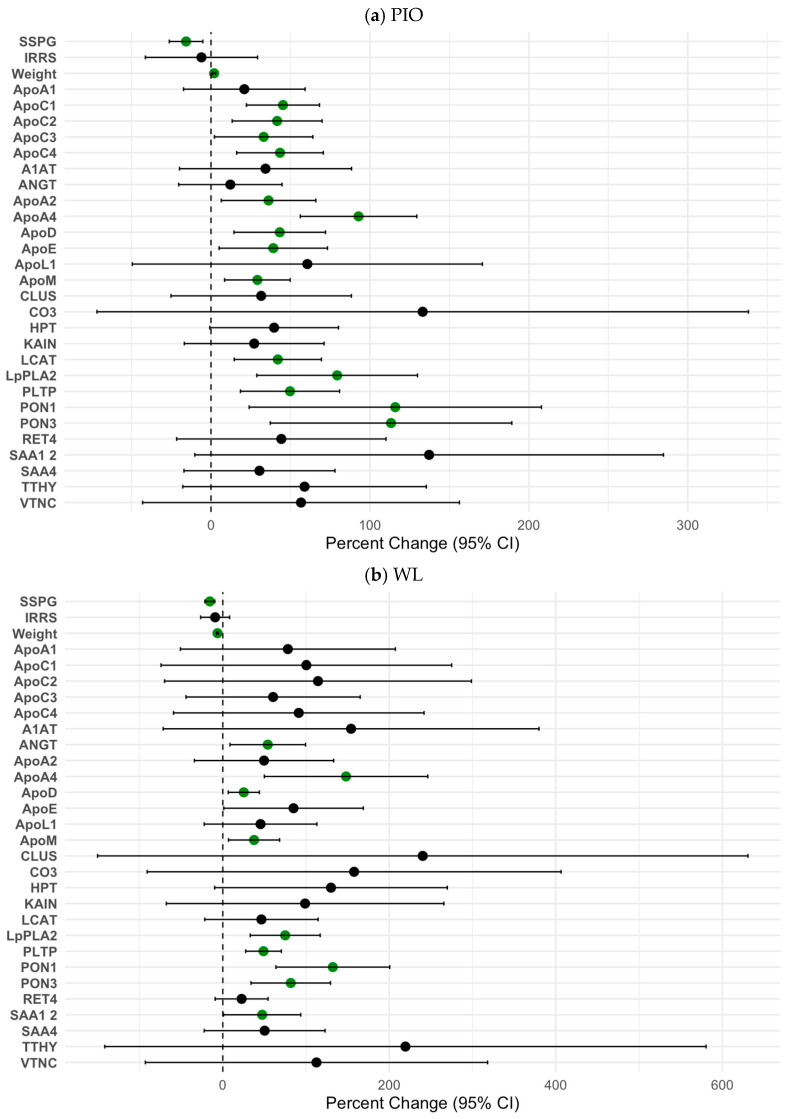
Percent Changes in IR (SSPG Concentration) and ApoA1 Proteins After PIO (**a**) and WL (**b**). SSPG indicates steady-state plasma glucose and is a direct measure of insulin resistance (IR) obtained during the insulin suppression test. A higher SSPG concentration indicates greater IR. Filled circles represent the percent change from baseline, and horizontal error bars represent the 95% confidence intervals of the percent change. Green circles represent a statistically significant association (*p* < 0.05).

**Table 1 ijms-26-10690-t001:** Baseline patient characteristics of the full cohort (N = 861).

Age, years	47.3 (11.6)
Female *	564 (65.5)
Non-Hispanic White *	590 (68.5)
Hispanic *	64 (7.4)
Black *	47 (5.5)
South Asian *	83 (9.6)
East Asian *	70 (8.1)
American Indian and Alaska Native *	7 (0.8)
Weight, kg	85.6 (19.6)
BMI	30.2 (6.0)
Total Cholesterol, mg/dL	189.9 (39.5)
Triglycerides, mg/dL	140.5 (128.5)
LDL-C, mg/dL	116.3 (31.6)
HDL-C, mg/dL	46.6 (13.4)
Fasting Glucose, mg/dL	96.8 (9.7)
Fasting Insulin, mU/L	10.0 (7.6)
SSPG, mg/dL	164.1 (73.4)

Unless noted, all values are reported as mean (SD). * Reported as *n* (%). BMI, body mass index (reported as kg/m^2^); LDL-C, low-density lipoprotein cholesterol; HDL-C, high-density lipoprotein; SSPG, steady-state plasma glucose.

**Table 2 ijms-26-10690-t002:** The Relationship of the Gold-Standard Measure of Insulin Resistance (SSPG Concentration) with Indirect Measures of Insulin Resistance.

Variable	N	r (95% CI) *	*p* Value
IRRS	861	0.73 (0.69–0.76)	<0.001
HOMA-IR	861	0.67 (0.63–0.71)	<0.001
TG/HDL-C	662	0.41 (0.34–0.47)	<0.001
TG	662	0.35 (0.28–0.41)	<0.001
HDL-C	662	−0.37 (−0.43–−0.30)	<0.001

* R values represent the Pearson correlation coefficients, and CI represents the 95% confidence intervals of the correlation coefficients. CI, confidence interval; IRRS, insulin resistance risk score; Homeostasis Assessment of Insulin Resistance, HOMA-IR; TG, triglycerides; HDL-C, high-density lipoprotein.

**Table 3 ijms-26-10690-t003:** Baseline patient characteristics of Interventional Cohort.

	PIO N = 38	WL N = 70	*p* Value
Age, years *	50.3 (8.1)	49.0 (10.3)	0.50
Female	13 (34.2)	22 (31.4)	0.94
Non-Hispanic White	25 (65.8)	36 (51.4)	0.283
Hispanic	2 (5.3)	5 (7.1)	0.82
Black	7 (18.4)	1 (1.4)	0.001
South Asian	1 (2.6)	20 (28.6)	0.001
East Asian	3 (7.9)	7 (10)	0.99
Am. Indian and Alaska Native	0 (0)	1 (1.4)	0.99

Unless noted, all values reported as n (%). * Reported as mean (SD); *p* value reflects unpaired *t* testing between PIO and WL cohorts.

## Data Availability

The data presented in this study are available on request from the corresponding author due to patient privacy reasons.

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
