# Peer review of "Changes in Apolipoprotein A1-Associated Proteomic Composition After Pioglitazone Treatment Versus Weight Loss"

_ijms, 2025, doi:10.3390/ijms262110690_

Round 1

Reviewer 1 Report

Comments and Suggestions for Authors

Title of manuscript: Changes in Apolipoprotein A1-Associated Proteomic Composition After Pioglitazone Treatment versus Weight Loss

Thank you for an intriguing and possibly significant study, authors. I offer a thorough, practical, and in-depth analysis of every text part beneath.

Major Comments:

Abstract

 Please include quantitative results in the Abstract: absolute/percent changes and exact p-values or FDR-adjusted p (e.g., “PON1 increased by X% (p=0.003, FDR=0.02)”).  Clarify the cohort/intervention sizes in one line (N total; PIO n=38, WL n=70). Replace vague phrases (“improved this profile”) with specific outcomes (which proteins changed, direction).

 Introduction

Although the beginning is well-written, there is repetition about For example, "We hypothesize that higher SSPG correlates with a more atherogenic ApoA-I proteome and that insulin-sensitizing interventions will partially reverse those alterations," would be a clear goal and testable hypothesis in one line.

Give a clearer explanation of the hypothesized mechanistic differences (PPAR-γ mediated transcriptional remodeling vs. adipose mass reduction) and why Pio Glitazone was chosen as the contrast instead of weight loss. Provide two or four current references if you haven't already, and cite the most recent proteomic research (2022–2025) that links the HDL proteome to function.

Methods

  1. Explain intervention assignment: Were individuals chosen or recruited independently, or were they randomized to PIO vs. WL?

Provide a CONSORT/flow diagram with the numbers of people that were screened, enrolled, and finished for each arm, along with the precise selection criteria for inclusion in each arm.

Addressing potential selection bias is necessary.

Give the dates of recruitment and the IRB permission number. Although IRB permission is stated in the manuscript, the approval ID is not displayed. The approval reference is required by journals.

Inconsistent Ns have been reported (861, 860, 858). Please reconcile and show a flow diagram listing the final N used for each analysis and reasons for exclusions / missing proteomic or SSPG data.

  1. Although you accurately explain SSPG in Methods, IRRS is presented in Table/Scheme using units (mg/dL). IRRS is a probability/score, hence this is untrue. Please correct units and labeling throughout the manuscript. Provide the full reference for the insulin suppression test protocol used (or a brief justification if rates differ from standard), and state whether the same technicians/labs processed all tests. Show the distribution (histogram) of SSPG values and justify the cutoff SSPG >132.5 mg/dL used to define IR (was median chosen a priori?). If chosen post-hoc, state that and consider alternative sensitivity analyses.

3-Provide specific technical parameters for the MS experiment, such as sample preparation, LC gradient, instrument model, acquisition settings, database/search engine, affinity enrichment technique with vendor/catalog of recombinant ApoA-I, and FDR for peptide/protein identification. To ensure reproducibility, this is crucial. You report CVs of "5-14%"; could you kindly supply per-protein CVs (Supplemental Table), the number of technical duplicates, the assay's dynamic range, and the limit of detection? Was the 3-month and baseline samples run in the same batches, according to the batch structure? How were potential batch effects (run order randomization, normalization, and QC samples) addressed?

Give proof that the results were not influenced by batch effects. Indicate if the reported protein quantification is absolute or relative to isotopic standards, and include the units (e.g., normalized intensity, ng/mL). Indicate the normalization technique if normalized intensities were used.Describe the number of ApoA-I-associated proteins found overall and the requirements for inclusion in targeted studies in the case of an untargeted finding. List all of the proteins found in the Supplementary Material

4-  The primary proteomics analyses used paired t-tests per protein. For repeated measures proteomic data with covariates and multiple proteins, I recommend linear mixed-effects models (random subject intercept) to account for within-subject correlation and to adjust for covariates (age, sex, baseline BMI, baseline protein abundance, and weight change). Please re-analyze key findings using such models and report adjusted effects (coefficients, 95% CI, p-values).

 You state Benjamini–Hochberg FDR was used at 5% — please report FDR-adjusted p-values for all proteomic comparisons in the main or Supplementary Tables (not only unadjusted p). Which results survive FDR?

ANCOVA (post vs. pre with baseline as a covariate) or mixed models with group×time interaction should be used instead of straightforward independent t-tests for comparing PIO and WL. Give the treatment*time interaction p-value for every protein.

 Examine the assumptions of normalcy. Use either log-transformation or non-parametric paired testing if protein distributions are skewed; specify which method was applied. P-values by themselves are not enough; include effect sizes (Cohen's d or percent change with 95% CI) for the primary proteomic results.  The method used to impute missing proteome values should be explained, and a sensitivity analysis comparing results with and without imputation should be included.

Results

Could you elaborate on the baseline distinctions between the WL and PIO groups and how these might impact results? Although the charts are instructive, you might want to include confidence intervals to better show variability. Certain proteome alterations, such as those in the pCAD index, were not statistically significant. What is the proper interpretation of these? Could you elaborate on whether the effects of the intervention on the ApoA1 proteome were altered by sex or ethnicity?

Discussion

Could you provide more details about the potential direct effects of elevated ApoC1–C4, ApoM, and PON1/3 on HDL function and ASCVD risk? The discussion highlights the wider impact of PIO; could you explain why, even while WL improved insulin resistance, its effects were less pronounced? Could you talk about if the detected proteome changes are likely to be temporary (3 months) or whether they could last longer? What are the clinical ramifications for patient treatment, and should future risk stratification take ApoA1 proteomic profiling into account? Could you elaborate on the limitations (possible selection bias, lack of long-term results, small PIO cohort)?

Conclusion

What implications can these findings have for the creation of biomarkers or future clinical trials?

Minor Comments:

Results

 Clarify the race/ethnicity categories and consider adjusted analyses if race/ethnicity modifies proteomic associations.

 Consider reporting partial correlations adjusted for age, sex, and BMI — these will show whether associations are independent of adiposity.

 In Supplementary Table 2, provide the exact p and FDR values for each protein together with the per-group mean (SD) pre and post values.

Investigate whether proteome alterations mediate the effect of intervention on lipid improvements for proteins that alter and for lipid outcomes (e.g., TG).  Subgroup analyses: considering the cohort composition (65.5% female, ~68% White), look at effects unique to sex and race/ethnicity. If underpowered, make sure to indicate it clearly.

clear legends that state the statistical test used, whether p values are FDR-adjusted, and what “green circles” or outlines denote. In Tables, report exact p-values (not only “<0.01”) and 95% CIs where possible

Discussion

(Extend the mechanistic discussion of how PPAR-γ activation by PIO may remodel the HDL proteome (liver/adipose gene expression, altered lipase activity, changes in lipoprotein exchange). However, keep in mind that a change in the proteome does not always translate into better HDL function; future research should measure cholesterol efflux and enzymatic activities.
 Include a paragraph about the clinical implications and cautions. What functional tests, longitudinal outcome studies, and pharmacological safety considerations would be required to convert proteomic remodeling into fewer ASCVD events?

Final Recommendation

The manuscript is scientifically sound and potentially publishable, but requires major revisions to improve clarity, interpretation, and generalizability.

Reviewer 2 Report

Comments and Suggestions for Authors

This article innovatively reports that the data indicate IR is linked to a pro-atherogenic ApoA1 proteome, and both interventions ameliorate this profile, with PIO exerting a broader proteomic influence. These results underscore the potential of ApoA1 proteome–targeted strategies to mitigate residual ASCVD risk. The experimental design is sound, and the arguments are well supported.

Reviewer 3 Report

Comments and Suggestions for Authors

This study demonstrated that both pioglitazone (PIO), a peroxisome proliferator-activated receptor gamma (PPARγ) agonist, and weight loss (WL) interventions exerted a potent impact on ApoA1-associated pathways. PIO significantly altered 14 proteins, including key lipid metabolism regulators like lecithin-cholesterol acyltransferase (LCAT), an enzyme critical for cholesterol esterification and the maturation of high-density lipoprotein (HDL) particles, and apolipoprotein E (ApoE), a pleiotropic apolipoprotein instrumental in mediating the hepatic uptake of triglyceride-rich lipoproteins and facilitating cholesterol efflux, thereby directly contributing to improved reverse cholesterol transport and a mitigated risk of atherosclerosis. These findings suggest PIO’s capacity for enhanced lipid homeostasis and potent anti-inflammatory effects, potentially conferring greater cardiovascular protection in insulin-resistant individuals. WL also demonstrated beneficial effects, notably by upregulating apolipoprotein M (ApoM), which supports endothelial function and anti-inflammatory responses, while promoting sustainable, root-cause metabolic improvements.

While the paper overall can be recommended for publication, several crucial points need clarification to enhance its robustness and address potential confounders:

  1. The significant 24-hour variability inherent in many lipid parameters, as documented in the literature (e.g., doi: 10.1111/jpi.12752), underscores the critical importance of sample collection timing. Furthermore, environmental light conditions and its seasonality have been shown to influence lipid metabolism and its circadian phasing (e.g., doi: 10.3390/biology14070799; doi: 10.1111/jpi.70023). Consequently, rigorous standardisation of measurement timing is essential. Please clarify whether blood samples for protein and SSG analysis were collected at the same time of day for all participants. Without such uniformity, potential confounding effects from circadian variations in proteins like ApoA1 or PON1/3 (which exhibit diurnal fluctuations of 10-30%, often peaking in the morning) may compromise the validity of the results. If sample collection times were not uniform, please provide the range of collection times or consider re-analysing the data with time-of-day as a covariate.

  2. Clarification is required regarding the seasons during which the pre- and post-intervention phases were conducted. The stated 3-month lag between phases (P.6, L.180) suggests that habitual environmental conditions may have undergone considerable changes. This seasonal variation could significantly impact the outcomes (cf. see above). The potential influence of these changing environmental conditions on the study’s findings should be addressed in the Discussion section.

  3. Furthermore, please consider that individuals with metabolic disorders and diabetes may be predisposed to co-factors associated with eveningness and excessive light-at-night (LAN) exposure (e.g. doi: 10.1093/sleep/zsac130). These factors may be plausibly linked to differential changes in apolipoprotein A1-associated proteomic composition. The potential interplay between metabolic status, chronotype, LAN excess and observed proteomic alterations should be discussed to provide a more comprehensive interpretation of the results.

  4. Please elucidate on the WL program's specifics (e.g., caloric deficit, meal timing, exercise). Changes in 24-hour eating patterns can shift circadian lipid metabolism (e.g., ApoE/ApoC3 rhythms), potentially influencing protein changes independently of weight reduction. Discuss this in the context of hepatic PPAR-gamma expression.

  5. PIO's effects on timing of lipid metabolism: please enhance the discussion on whether PIO alters circadian lipid metabolism timing. As a PPAR-gamma agonist, it may modulate clock genes (e.g., BMAL1), shifting peaks in LCAT or ApoE. Compare to WL and link to chronotherapy, such as optimal morning dosing.

Round 2

Reviewer 1 Report

Comments and Suggestions for Authors

The authors took these comments into account and, where technically feasible, conducted additional experiments.

The manuscript may be accepted for publication.

Reviewer 3 Report

Comments and Suggestions for Authors

I thank the authors for addressing my concerns. I have no further comments at this time.